# Towards Anytime Fine-tuning: Continually Pre-trained Language Models with Hypernetwork Prompts

**Gangwei Jiang[1]\*, Caigao Jiang[2], Siqiao Xue[2], James Y. Zhang[2],**
**Jun Zhou[2], Defu Lian[1], Ying Wei[3]†**

[1]University of Science and Technology of China , [2]Ant Group
[3]Nanyang Technological University

gwjiang@mail.ustc.edu.cn, caigao.jcg, siqiao.xsq, james.z@antgroup.com,
jun.zhoujun@antgroup.com, liandefu@ustc.edu.cn, ying.wei@ntu.edu.sg

## Abstract

Continual pre-training has been urgent for adapting a pre-trained model to a multitude of domains and tasks in the fast-evolving world. In practice, a continually pre-trained model is expected to demonstrate not only greater capacity when fine-tuned on pre-trained domains but also a non-decreasing performance on unseen ones. In this work, we first investigate such anytime fine-tuning effectiveness of existing continual pre-training approaches, concluding with unanimously decreased performance on unseen domains. To this end, we propose a prompt-guided continual pre-training method, where we train a hypernetwork to generate domain-specific prompts by both agreement and disagreement losses. The agreement loss maximally preserves the generalization of a pre-trained model to new domains, and the disagreement one guards the exclusiveness of the generated hidden states for each domain. Remarkably, prompts by the hypernetwork alleviate the domain identity when fine-tuning and promote knowledge transfer across domains. Our method achieved improvements of 3.57% and 3.4% on two real-world datasets (including domain shift and temporal shift), respectively, demonstrating its efficacy.

## 1 Introduction

Pre-trained language models (LMs), such as GPT-3 (Brown et al., 2020) and BERT (Devlin et al., 2019a), have revolutionized a wide spectrum of downstream natural language processing (NLP) tasks. Being initially pre-trained on a vast unlabeled corpus (e.g., $C_0$ in Fig. 1), unfortunately, they struggle to keep up to date with language evolution (e.g., *emerging internet slang, expanded meaning of "Omicron"*) and domain shift (e.g., *electronic health records for medical diagnosis*).

---

\* This work was done when the author Gangwei Jiang was at Ant Group for intern.

† Corresponding author.

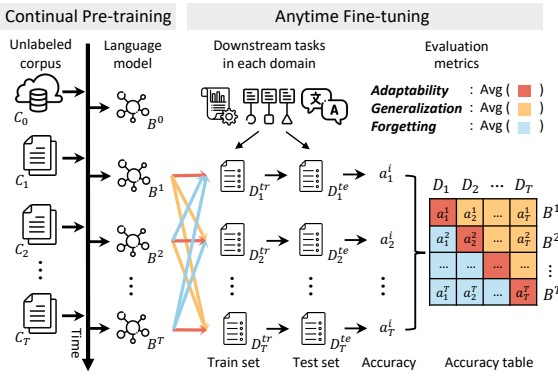

Figure 1: Illustration of continual pre-training and the evaluation protocol of anytime fine-tuning, in which $a_j^i$ in the accuracy table denotes the fine-tuned accuracy of the LM at any $i$-th stage, i.e., $B^i$, on the $j$-th *pre-trained* (blue), *current* (red), and *unseen* domains (orange).

Continual pre-training methods (Jin et al., 2022; Ke et al., 2023) have recently emerged to address it by continually adapting an LM to a sequence of domains (e.g., $T$ domains in Fig. 1). Two major lines of existing approaches, including knowledge distillation (Jin et al., 2022) and parameter isolation (Ke et al., 2023, 2022a), make strides toward (1) maximizing the *adaptability*, i.e., the performance of an LM (e.g., $B^2$ in Fig. 1) when fine-tuning it onto the domain where it is pre-trained (e.g., $D_2$ in Fig. 1), and (2) avoiding *catastrophic forgetting* (CF), which is measured by the fine-tuned performance of an LM (e.g., $B^2$ in Fig. 1) on the already pre-trained domains (e.g., $D_1$ in Fig. 1).

Beyond the above two criteria, in practice, a continually pre-trained LM is also anticipated to offer non-decreasing *generalization* capability on unseen domains. As illustrated in Fig. 1, it is likely that the unlabeled corpus for the domain of interest (e.g., electronic health records as $D_T$) remains inaccessible to an LM (e.g., $B^2$) beforehand, while this LM should be superior or at least on par with its preceding models (e.g., $B^1$) on the $T$-th domain. On

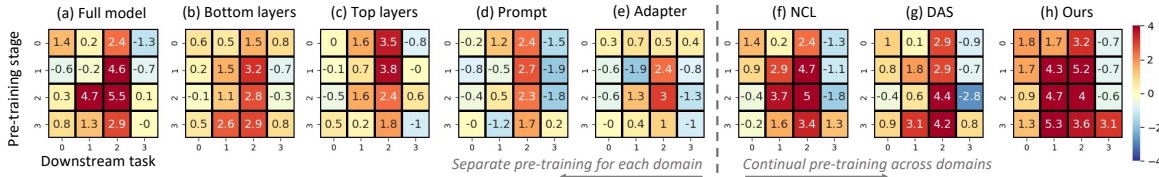

Figure 2: Evaluation of separate and continual pre-training methods under anytime fine-tuning, where we modify each value $a_j^i$ by subtracting $a_j^0$ as the fine-tuned accuracy of the initial LM $B^0$. (a)-(e) show the accuracy tables by pre-training each domain separately *w.r.t.* different sets of parameters (e.g., top layers); (f)-(h) are by the naively continual pre-training method (NCL), DAS (Ke et al., 2023), and ours. Detailed settings are available in Sec. 5.2.

this account, we propose the comprehensive evaluation protocol named ***anytime fine-tuning*** that subsumes all the three aspects, where a continually pre-trained LM can be fine-tuned and evaluated on either previously pre-trained, current, or unseen domains. The effectiveness of current methods in terms of anytime fine-tuning remains largely unclear.

In this paper, we first conduct an empirical investigation of existing pre-training approaches under anytime fine-tuning (see Fig. 2) and identify the following two prominent unresolved research questions. **(1)** Parameter-efficient pre-training, such as training adapters (Ke et al., 2021b) and prompts (Razdaibiedina et al., 2023; Smith et al., 2023) only for each individual domain, does not even contribute greater *adaptability* than that before pre-training (i.e., evidenced in negative diagonal values of Fig. 2(d)(e)). Likewise, pre-training parts of parameters for each domain, may also diminish adaptability, through comparison of Fig. 2(b)(c)(g) with (a). **(2)** Continual pre-training is likely at the cost of sacrificing *generalization* to unseen domains, shown by large negative values in the third column of Fig. 2(f)(g).

To address the above issues, we propose a **H**ypernetwork **P**rompt guided **C**ontinual **P**re-**T**raining method (namely HPrompt-CPT[1]) that strikes a balance between forgetting, adaptability, and generalization. *First,* inspired by recent success of prompt engineering paired with full fine-tuning in domain adaptation (Radford et al., 2019; Brown et al., 2020), we introduce the hnet-prompt module consisting of a hypernetwork to automatically generate domain-specific prompts without handcrafted engineering. Different from parameter-efficient pre-training that train prompts only, we optimize both the hypernetwork and the full LM so

as to fully adapt to the current domain. An added benefit of hypernetwork prompts is that they eliminate the reliance on the domain identity to pinpoint prompts when fine-tuning. *Second*, we maximally preserve the generalization while mitigating CF of a continually pre-trained LM via the agreement and disagreement losses. We prompt the previous and current LM with a random prompt that simulates generic or learned domains and introduce the agreement loss to enforce consistency between their predictions to avoid forgetting while preserving model plasticity on other prompts. On the other hand, the disagreement loss promotes the exclusiveness of generated hidden states for the current domain, thus minimizing interference to the established knowledge and encouraging generalization during fine-tuning through diverse domain knowledge. Noteworthy, the hypernetwork also favors knowledge generalization, compared to disparate prompts of different domains.

**Main Findings and Contributions. (1)** We establish a continual pre-training evaluation protocol, called anytime fine-tuning, and empirically verify that existing parameter-efficient approaches lose their competitive edge in adaptability and almost all methods are at risk of impairing generalization to unseen domains (see Fig. 2). **(2)** We further conquer the two challenges by proposing a hypernetwork prompt guided continual pre-training (HPrompt-CPT) scheme where we train the hypernetwork with both the agreement and disagreement losses. HPrompt-CPT is effective, achieving the state-of-the-art on two real-world datasets.

## 2 Related Work

Continual Learning (CL) focuses on the problem of sequential learning from a stream of data that comes in different distributions. It has achieve a great success in computer vision (Wang et al., 2022a,c; Smith et al., 2023), natural language pro-

---

[1]The code of HPrompt-CPT will be released at https://github.com/gangwJiang/HPrompt-CPT

cessing (Sun et al., 2019; Ke et al., 2023), and data mining (Hao et al., 2023; Xue et al., 2023). In this paper, we focus on one of the important aspects, continual pre-training and present recent progresses below. More related works are given in Appendix A.

**Continual Pre-training.** Previous studies (Gururangan et al., 2020; Dery et al., 2022) have demonstrated that the fine-tuned performance of LM on downstream tasks can be enhanced by continued training on a domain-related corpus. Recent works take this concept further by introducing *Continual Pre-training* (CPT), where LM continually learns from streaming domain corpora. Jin et al. (2022); Jang et al. (2022) investigate conventional CL methods in CPT using real-world datasets and highlight the final LM can be fine-tuned to serve any task in pre-trained domains, leading to improved performance, while (Hu et al., 2022a) finds CPT is comparable with joint pre-training. To improve upon this, ELLE (Qin et al., 2022) progressively expands LMs with function-preserving initialization to inject knowledge from new corpus, while CPT (Ke et al., 2022a) designs specific adapters and utilizes a hard-masking to avoid CF. Additionally, DGA (Ke et al., 2022b) and DAS (Ke et al., 2023) adopt soft-masking to directly controls the update of the entire LM and contrast the previous and current representations.

Though these methods alleviate CF during CPT, they ignore the importance of adaptation to domain knowledge for better fine-tuned performance (Gururangan et al., 2020; Dery et al., 2022) and generalization to unseen domains (Wortsman et al., 2022; Andreassen et al., 2022). Our work utilizes the potential of LM and improves all three aspects.

## 3 Preliminaries

Our language model $B$ is constructed using the Roberta architecture (Liu et al., 2019), which is based on a bi-directional Transformer structure. LM takes a text sentence $\mathbf{x}_{1:T} = [x_1, x_2, ..., x_T]$ as input and encodes it into a contextual embedding $\mathbf{h} = [h_1, h_2, ..., h_T] = B(\mathbf{x}_{1:T})$.

### 3.1 Pre-training and Fine-tuning Tasks

During pre-training, the model is trained to predict missing words in a given text sentence $\mathbf{x}$ and thus acquires a general understanding of languages, such as syntax, semantics, and context. The pre-training task is called masked language modeling (MLM) (Devlin et al., 2019a), and the objective is $\ell_{mlm}(\mathbf{x}, \mathcal{W}) = -\sum_{\hat{x} \in m(\mathbf{x})} \log p\left(\hat{x} \mid \mathbf{x}_{\backslash m(\mathbf{x})}, \mathcal{W}\right)$, where $\mathcal{W}$ denotes the parameters of language model $B$, $m(\mathbf{x})$ and $\mathbf{x}_{\backslash m(\mathbf{x})}$ the masked words from $\mathbf{x}$ and the remain words, respectively. The conditional probability is calculated by a prediction layer $g_{mlm}$ as $p\left(\hat{x} \mid \mathbf{x}_{\backslash m(\mathbf{x})}, \mathcal{W}\right) = g_{mlm}\left(B_{\mathcal{W}}(\mathbf{x}_{\backslash m(\mathbf{x})})\right)$.

After pre-training, the model is fine-tuned using a smaller dataset specific to a downstream task, which enables it to learn the intricacies and details of the task. In our study, the downstream task contains labeled samples $(\mathbf{x}, y)$ (e.g., in a hashtag prediction task, $\mathbf{x}$ is the user's twitter and $y$ is the selected hashtag). Its objective function is to minimize $\ell_{down}(\mathbf{x}, \mathcal{W}) = -\log p\left(y \mid \mathbf{x}, \mathcal{W}\right)$.

### 3.2 Soft Prompt Learning

Prompt tuning (Lester et al., 2021) is a lightweight alternative to the full fine-tuning that introduces a trainable prompt $\mathbf{P} = [p_1, p_2, ..., p_L]$ as a prefix to the input embedding $\mathbf{E} = [e(x_1), e(x_2), ..., e(x_T)]$ to replace the update on entire model. The prompt length is $L$, $e$ represents the embedding layer in LM, and $p_i \in \mathbb{R}^d$ has the same dimension $d$ as the token embedding. During prompt tuning, the concatenated matrix $[\mathbf{P}; \mathbf{E}] \in \mathbb{R}^{(L+T) \times d}$ is used as the input to the LM, expressed as $B(\mathbf{x}, \mathbf{P})$. The downstream task optimization is represented as $\ell_{down}(\mathbf{x}, \mathbf{P}) = -\log p\left(y \mid \mathbf{x}, \mathbf{P}\right) = -\log g_{down}\left(B(\mathbf{x}, \mathbf{P})\right)$, where $g_{down}$ is the prediction layer for the task and the model $B$ does not update in conventional soft prompt learning.

### 3.3 Continual Pre-training for Anytime Fine-tuning

Continual pre-training (Jang et al., 2022; Meng et al., 2023) is a way to efficiently adapt to the new domain while maintaining learned knowledge. The problem formulation is as follows (see Fig. 1): assume a stream of new domains (e.g., *latest news about "Omicron"*) sequentially appears as $\mathcal{D}_1, ..., \mathcal{D}_N$, where $\mathcal{D}_i$ is the distribution of $i$-th domain over a finite vocabulary of tokens $\mathcal{X}$. Initially, we have an LM that has been well pre-trained on the general corpus $C_0$, such as Roberta. Then at each stage $i$, a collection of new unlabeled corpus $C_i = \{\mathbf{x} \mid \mathbf{x} \in \mathcal{D}_i\}$ is obtained. The existing LM continually pre-trains to learn the new knowledge from $\mathcal{D}_i$, with the goal of improving performance for ***anytime fine-tuning***, where the LM is expected

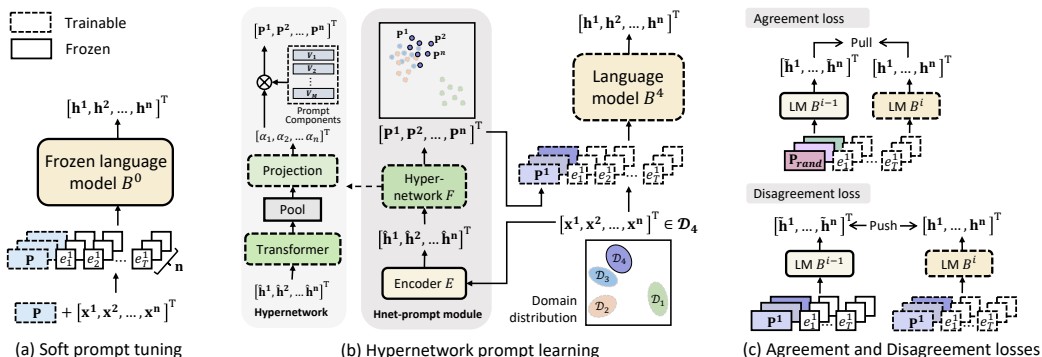

Figure 3: An overview of the model structure, with dotted lines indicating trainable modules and solid lines indicating frozen modules. (a) denotes the soft prompt tuning (Sec. 3.2). (b) shows the pre-training on domain 4 with the hnet-prompt module (Sec. 4.1). The hypernetwork takes the contextual embedding $\hat{h}$ as input and automatically generates a prompt $\mathbf{P}$ considering domain and sample properties, which clusters $\mathbf{P}$ for similar domains ($\mathcal{D}_2,\mathcal{D}_3,\mathcal{D}_4$) together and facilitates knowledge generalization. (c) computes the agreement and disagreement losses (Sec. 4.2).

to get greater capacity when fine-tuned on tasks from all pre-trained, current, and unseen domains.

Each domain has its labeled dataset $D_i = \{(\mathbf{x}, y) \mid y = F^*(\mathbf{x}), \mathbf{x} \in \mathcal{D}_i\}$, where $F^* \in \mathcal{Y}$ provides ground-truth labels for classification. During the evaluation, the LM $B^i$, pre-trained up to the $i$-th domain, is fine-tuned on a train set $D_j^{tr}$ and then tested on $D_j^{te}$ to measure its domain performance, as illustrated in Fig. 1. The resulting accuracy, denoted as $Acc_{D_j}^{B^i}$ (simplified as $a_j^i$), indicates the model capacity on task $D_j$ as well as the degree of knowledge of $j$-th domain maintained by LM after being sequentially trained up to $C_i$.

Through the integration of results, an accuracy table is generated, allowing for the computation of three crucial metrics in anytime fine-tuning as discussed in Sec. 1: adaptability, generalization, and forgetting. The values used to calculate these metrics are indicated by different colors in Fig. 1. Red cells along the diagonal of the table represent adaptability, indicating the degree to which the LM learns knowledge relevant to current domain. Yellow cells in the upper triangle represent generalization, signifying the ability to perform effectively in future domains. Blue cells in the lower triangle represent forgetting, reflecting a reduction in previously learned knowledge during training.

## 4 Method

A successful algorithm of continual pre-training for anytime fine-tuning should meet the following requirements: (1) effective adaptation to the current domain and capturing more domain knowledge, (2) strong generalization to tasks in unseen domains, and (3) minimal catastrophic forgetting of previously learned knowledge. To achieve this, we propose a framework, dubbed HPrompt-CPT, which consists of two components: the *Hnet-Prompt* module and *Agreement* and *Disagreement* losses. The overview is presented in Fig. 3.

### 4.1 Hnet-Prompt for Pre-training and Fine-tuning

Previous soft prompt methods (Qin and Joty, 2022; Zhu et al., 2022; Razdaibiedina et al., 2023) have made great success in the CL, with almost no catastrophic forgetting. However, these parameter-efficient methods fall short in model adaptation during the pre-training stage and fail to exhibit generalization capabilities when faced with new domains, as shown in Fig. 2. On the other hand, prompt engineering has shown exceptional performance in pre-training language models to better learn domain-specific knowledge (Radford et al., 2019; Brown et al., 2020). However, the use of hard-coded prompts makes it difficult to implement and less relevant to generalization.

Therefore, inspired by previous meta-learning approaches (Qiao et al., 2018; Yao et al., 2019), we propose a prompt module with a meta hypernetwork (Hnet-Prompt) for automatic knowledge adaptation and cross-domain generalization. Specifically, when a batch of data $[\mathbf{x}^1, ..., \mathbf{x}^n]$ in a specific domain $\mathcal{D}_i$ comes, the hypernetwork generates a prompt $\mathbf{P}$ for each sample (see Fig. 3(b)), taking into account both domain and sample properties while generalizing knowledge from learned domains. The process is parameterized as:

$$\mathbf{P}^i = F(\hat{\mathbf{h}}^i) = F(E(\mathbf{x}^i)), \quad (1)$$

where $E$ refers to a text encoder, $F$ corresponds to a hypernetwork, and $\hat{\mathbf{h}}^i$ represents the contextual embedding, which captures both the sentence and implicit domain information.

Hypernetwork $F$ encodes the domain feature of input samples (we use a 6-layer Transformer) and then projects the pooled feature to obtain the prompt (see Fig. 3(b)). Rather than directly generating the prompt, we set $M$ prompt components $\mathbf{V}_m \in \mathbb{R}^{L \times d}$ and generate a weight vector $\alpha \in \mathbb{R}^M$ to get the final prompt $\mathbf{P} = \sum_{m=1}^M \alpha_m \mathbf{V}_m$. Vector $\alpha$ controls the contribution of each prompt component, which corresponds to a basic domain. This approach reduces the parameter of the linear layer for projection and alleviates forgetting by shifting the learning problem from remembering the entire embedding to a weight vector.

Prompt components $\mathbf{V}$, analogous to a set of basis vectors, are a set of prompt embeddings that are randomly initialized, trainable and optimized through gradient descent. The well-trained prompt components are supposed to offer greater generalization to future domains as long as the prompt components are as mutually exclusive as possible. For example, a prompt embedding directly optimized for the domain of "ACL papers" does not directly apply to the domain of "AI papers" due to the domain difference; however, one of the prompt components learned on "ACL papers", e.g., "deep learning", can be combined with another component of "statistics" to generalize to the domain of "AI papers".

During pre-training, the language model is conditioned on the prompt generated by the hypernetwork, which models $p(output \mid input, domain)$ and injects the domain knowledge into the model in an explicit way. Then, we optimize the language model and hypernetwork in an end-to-end manner by minimizing the following equation:

$$\ell_{mlm}(\mathbf{x}, \mathcal{W}, \Theta) = \\ - \sum_{\hat{x} \in m(\mathbf{x})} \log p\left(\hat{x} \mid \mathbf{x}_{\backslash m(\mathbf{x})}, \mathcal{W}, \Theta\right), \quad (2)$$

where $p(\cdot) = g_{mlm}\left(B_{\mathcal{W}}\left(\mathbf{x}_{\backslash m(\mathbf{x})}, F_{\Theta}\left(\mathbf{x}_{\backslash m(\mathbf{x})}\right)\right)\right)$ and $\Theta$ is the parameter of $F$. This approach allows for qualified and automatic adaptation to domain knowledge and enables the transfer of this knowledge across domains through hypernetwork.

During downstream task fine-tuning, domain identity is not required anymore. Hypernetwork will automatically map the input samples to

their unique prompt embedding with the knowledge generalized from learned domains. Given a task $t$, the entire model will be fine-tuned on the smaller labeled dataset, using the objective $\ell_{down}(\mathbf{x}, \mathcal{W}, \Theta) = -\log p(y \mid \mathbf{x}, \mathcal{W}, \Theta)$. Here hypernetwork $F$ is also trainable to get the best adaptation to downstream tasks. The fine-tuned performance on the task shows the degree of domain knowledge maintained by the LM.

## 4.2 Agreement and Disagreement Losses for Prompted Language Model

While preventing the forgetting of learned knowledge is always the key challenge in continual pre-training, they are at the cost of adaptability and generalization. To overcome it, we propose a novel approach, named agreement and disagreement losses.

**Agreement loss**. While knowledge distillation (KD) has been demonstrated to perform well in overcoming CF (Chuang et al., 2020; Dong et al., 2021), its alignment on the entire feature space can limit the adaptation to new domains. To alleviate it, we propose to align the output $p(output \mid input, domain)$ of the prompted language model instead $p(output \mid input)$ used in conventional KD. We term this approach the ***agreement loss***. Specifically, we begin with the prior learned LM $B^{i-1}$. Then, initialize the random prompt $\mathbf{P}_{rand}$ and generate prompted hidden states using both current LM $B^i$ and previous LM $B^{i-1}$ (see Fig. 3(c)). We then minimize the distance metrics $\mathcal{M}$ between the outputs of two models, as shown below:

$$\ell_a(\mathbf{x}, \mathcal{W}) = \mathcal{M}[B^{i-1}(\mathbf{x}, \mathbf{P}_{rand}), \\ B^i_{\mathcal{W}}(\mathbf{x}, \mathbf{P}_{rand})], \quad (3)$$

where $\mathbf{P}_{rand}$ simulates the condition to active generic or learned domain knowledge. The agreement loss, which operates on $B(\cdot, \mathbf{P}_{rand})$, effectively prevents forgetting by enforcing consistency on multiple randomized conditions and preserves the plasticity to new domains by maintaining model capacity conditioned on other prompts, as demonstrated by a comparison to KD. A smaller $\mathcal{M}$ indicates a closer distance between the two inputs. In this article, we use cosine similarity to calculate $\mathcal{M}$, which performs better than the KL distance between logits in the experiments in Sec. 5.4.

**Disagreement loss**. Besides the consistency achieved by agreement loss, we also expect the exclusiveness of the generated hidden states for the current domain. It brings two advantages: (1) it reduces interference to established knowledge, which

Table 1: Performance of baseline results on DAPset/TWEET benchmarks (all results reported in this paper are averaged over 4 random seeds). The symbol "−" in the table is because $F\_Acc$ is the same as the average accuracy $A\_Acc$ in the separate pre-training settings. *We also report the results for different domain orders in Appendix D.*

| Setting | Method | DAPset | | | TWEET | | |
|---|---|---|---|---|---|---|---|
| | | $A\_Acc$ | $O\_Acc$ | $F\_Acc$ | $A\_Acc$ | $O\_Acc$ | $F\_Acc$ |
| Separate Pre-training | Initial | $0.8053 \pm 0.010$ | $0.8171 \pm 0.010$ | - | $0.7933 \pm 0.001$ | $0.7935 \pm 0.001$ | - |
| | Multi-Task | $0.8203 \pm 0.002$ | $\mathbf{0.8299} \pm 0.005$ | - | $0.8014 \pm 0.002$ | $0.8047 \pm 0.001$ | - |
| | One-Full | $0.8235 \pm 0.007$ | $0.8174 \pm 0.008$ | - | $0.8037 \pm 0.001$ | $0.8064 \pm 0.001$ | - |
| | One-Adapter | $0.8060 \pm 0.008$ | $0.8172 \pm 0.003$ | - | $0.7913 \pm 0.002$ | $0.7915 \pm 0.003$ | - |
| | One-Prompt | $0.8101 \pm 0.012$ | $0.8109 \pm 0.012$ | - | $0.7873 \pm 0.002$ | $0.7876 \pm 0.002$ | - |
| Continual Pre-training | NCL | $0.8298 \pm 0.005$ | $0.8189 \pm 0.006$ | $0.8198 \pm 0.005$ | $0.8108 \pm 0.002$ | $0.8094 \pm 0.001$ | $0.8079 \pm 0.001$ |
| | EWC | $0.8082 \pm 0.004$ | $0.8109 \pm 0.003$ | $0.8020 \pm 0.003$ | $0.8028 \pm 0.001$ | $0.8048 \pm 0.001$ | $0.8037 \pm 0.001$ |
| | DERpp | $0.8245 \pm 0.002$ | $0.8174 \pm 0.004$ | $0.8239 \pm 0.001$ | $0.8102 \pm 0.001$ | $0.8087 \pm 0.001$ | $0.8118 \pm 0.001$ |
| | LwF | $0.8239 \pm 0.003$ | $0.8229 \pm 0.006$ | $0.8179 \pm 0.006$ | $0.8021 \pm 0.002$ | $0.7986 \pm 0.002$ | $0.8082 \pm 0.001$ |
| | CoDA-Prompt | $0.8141 \pm 0.002$ | $0.8161 \pm 0.004$ | $0.8176 \pm 0.004$ | $0.7931 \pm 0.001$ | $0.7954 \pm 0.001$ | $0.7958 \pm 0.001$ |
| | DAS | $0.8221 \pm 0.004$ | $0.8164 \pm 0.001$ | $0.8251 \pm 0.006$ | $0.8066 \pm 0.001$ | $0.8078 \pm 0.001$ | $0.8099 \pm 0.003$ |
| | Ours | $\mathbf{0.8356} \pm 0.002$ | $0.8277 \pm 0.003$ | $\mathbf{0.8341} \pm 0.003$ | $\mathbf{0.8186} \pm 0.001$ | $\mathbf{0.8168} \pm 0.002$ | $\mathbf{0.8203} \pm 0.001$ |

mitigates forgetting (Farajtabar et al., 2020; Wang et al., 2021b); (2) it encourages generalization when fine-tuning by incorporating a wider range of domain knowledge (Pagliardini et al., 2023). To achieve this exclusiveness, we add a loss function called ***disagreement loss***. Specifically, when a sample comes, we generate the prompt using hypernetwork $F$ and train the prompted LM to maximally disagree with the output of the previous LM, which is also promoted by the same embedding (see Fig. 3(c)). This involves minimizing the agreement metric $\mathcal{A}(\cdot, \cdot)$ to push apart the two prompted hidden states:

$$\ell_{da}(\mathbf{x}, \mathcal{W}, \Theta) = \mathcal{A}(B^{i-1}(\mathbf{x}, F(\mathbf{x})), \\ B^i_{\mathcal{W}}(\mathbf{x}, F_\Theta(\mathbf{x}))), \quad (4)$$

thereby increasing the exclusiveness of the output of LM for the current domain. In Sec. 5.4, we compare various implementation of $\mathcal{A}$ including orthogonal constrain (Smith et al., 2023), softmax variant (Pagliardini et al., 2023), opposite value of KL-divergence. Ultimately, we select the orthogonal constraint, which can be calculated using the equation $\mathcal{A}_{ortho}(\mathbf{X}, \mathbf{Y}) = ||\mathbf{X}\mathbf{Y}^T - \mathbf{I}||$.

Finally, the loss function of our HPrompt-CPT during pre-training can be summarized as follows:

$$\mathcal{L} = \sum_{i=1}^{N} \ell_{mlm} + \lambda_1 \ell_a + \lambda_2 \ell_{da}, \quad (5)$$

where $N$ is the batch size, and $\lambda_1, \lambda_2$ are the trade-off hyper-parameters. The loss input $\mathbf{x}_i$ is omitted.

## 5 Experiment

In this section, we conduct experiments on two benchmarks to investigate the adaptability, generalization, and degree of forgetting of HPrompt-CPT.

### 5.1 Benchmarks

**DAPset.** It is a benchmark for continual domain adaptive pre-training, originally constructed by (Ke et al., 2023). It consists of six domains, each with an unlabeled corpus and a corresponding end-task classification dataset. Each domain contains a corpus size of over 100 million tokens, and we follow the original data construction and task order.

**TWEET.** We develop a new benchmark based on a tweet dataset (Jin et al., 2022) to simulate the distribution shift over time. The dataset includes tweets from 2015 to 2019 and is split into five time periods to form five domain corpora, each with over 50 million tokens. The tweet texts are pre-processed following Nguyen et al. (2020). For the downstream task, we build a single-label hashtag prediction dataset for each domain following Gong and Zhang (2016). TWEET keeps the chronological order of domains to simulate the updating in the real-world system. Please refer to Appendix B for more information about the two benchmarks.

### 5.2 Metrics and Baselines

**Metrics.** We introduce three attributes of continual pre-training in Sec.3.3 and provide an explanation of their evaluation methods. Formally, we utilize the adaptation accuracy $A\_Acc = \frac{1}{T}\sum_{i=1}^{T} a_i^i$ to measure adaptability, the out-of-domain accuracy $O\_Acc = \frac{2}{T*(T-1)}\sum_{i=1}^{T}\sum_{j=i+1}^{T} a_j^i$ to evaluate generalization, and the final accuracy $F\_Acc = \frac{1}{T}\sum_{i=1}^{T} a_i^T$ to assess the degree of catastrophic forgetting. Here, $a_i^j$ represents the fine-tuned accuracy on the $i$-th downstream task, after being sequentially trained up to corpus $C_j$ in the $j$-th domain.

**Baselines.** We first evaluate the algorithms that build *separate model* for each domain, including: (1) **Initial** is fine-tuned on the initial

pre-trained point. (2) **Multi-Task** is domain-adaptively pre-trained on the mixture of all domains. (3) **One-Full** is domain-adaptively pre-trained with the updates on the full model. (4) **One-Adapter** is domain-adaptively pre-trained with an adapter layer (Houlsby et al., 2019). (5) **One-Prompt** is domain-adaptively pre-trained with a new prompt (Lester et al., 2021). Additionally, we test 7 *continual pre-training* methods: (6) **NCL** is sequentially pre-trained without any CL methods. (7) **EWC** (Kirkpatrick et al., 2017) is a regularization method that penalizes changes to important neurons. (8) **DERpp** (Buzzega et al., 2020) is a replay method in both sample and feature levels. (9) **LwF** (Li and Hoiem, 2017) uses knowledge distillation to protect previous predictions. (10) **CoDA-Prompt** (Smith et al., 2023) uses a set of prompt components to learn domain-specific knowledge. (11) **DAS** (Ke et al., 2023) is a parameter-isolation method which adopts soft-masking.

For HPrompt-CPT, we adopt a 6-layer Transformer as our hypernetwork and frozen Roberta as text encoder. We set the prompt length to 50, and the size of prompt components to 100. In addition, we implement a replay loss to the hypernetwork with a memory buffer storing 300 samples to get the best performance, while removing it resulting in a minimal drop of 0.24% in $F\_Acc$ on DAPset. During fine-tuning, we train each task for 15 epochs with an early stopping mechanism using the validation data (30% of testing data). We include additional ***Implementation Details*** in Appendix C.

### 5.3 Results and Analysis

**Comparison with the state-of-the-art.** Table 1 shows the continual pre-training performance of different methods on three dimensions. From these results, we make the following observations:

*Observation 1:* HPrompt-CPT outperforms baselines in terms of adaptability, generalization, and avoidance of catastrophic forgetting. Our approach achieves new state-of-the-art results across all three metrics, with increases of 1.38% and 1.09% on the DAPset in terms of generalization and final performance compared to the most recent algorithm, DAS, as depicted in the last row of Table 1. These results highlight the advantages of injecting domain knowledge into the LM with the hnet-prompt module, which aids in adaptation and promotes knowledge transfer.

*Observation 2:* Naive multi-task learning is

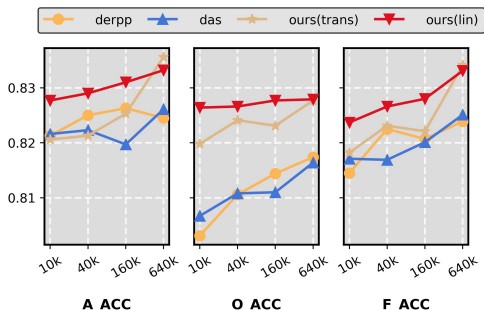

Figure 4: Performances on DAPset with different sizes of the corpus. The implementations of "ours (trans/lin)" refer to utilizing transformer/linear hypernetwork in HPrompt-CPT, respectively.

sub-optimal for continual pre-training. Our hnet-prompt method achieves a relative improvement in $F\_Acc$ of 1.69% on DAPset and 2.35% on TWEET, suggesting that it can alleviate negative transfer between conflicting domains and minimize forgetting. It is worth noting that the $O\_Acc$ metric of multi-task learning cannot be compared fairly with other algorithms since it has already observed all domains. Nevertheless, our algorithm still achieves a 1.50% gain on TWEET, which may result from the generalization of the diverse domain knowledge in HPrompt-CPT.

*Observation 3:* Full model tuning achieves better results in learning and transferring domain knowledge. Our proposed method and NCL outperform parameter-efficient methods such as One-Adapter, One-Prompt, and CoDA-Prompt. Interestingly, methods that incorporate regularization terms on parts of neurons, such as EWC and DAS, also result in lower $A\_Acc$. This suggests that injecting a large amount of domain knowledge into the LM requires a sufficient number of trainable parameters. Our prompted LM, with all parameters trainable and no empirical constraints on updates, shows the best adaptation performance.

**Data-efficient pre-training.** Note that we hypothesize that HPrompt-CPT is especially effective in the setting of anytime fine-tuning. Its performance on a small subset of the corpus is worth referring to, for the model can be utilized for fine-tuning in cases where a domain is not finished training. Fig. 4 illustrates the performances trained on different sizes of datasets and highlights the effectiveness of our method in low-resource environments, particularly in terms of generalization ability. Our design of the hnet-prompt module successfully promotes knowledge transfer across domains, and besides we ob-

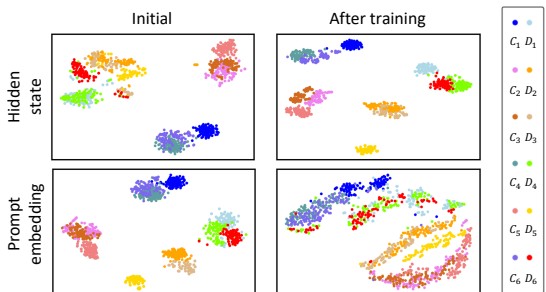

Figure 5: The t-sne map about prompt embedding and hidden state of the last layer. $C_i$ and $D_i$ denote the corpus and downstream task in $i$-th domain, respectively.

Table 2: Ablation results on the main components.

| Hypernetwork | $\ell_a$ | $\ell_{da}$ | $A\_Acc$ | $O\_Acc$ | $F\_Acc$ |
|:---:|:---:|:---:|:---:|:---:|:---:|
| ✗ | ✗ | ✗ | 0.8165 | 0.8066 | 0.8114 |
| ✗ | ✓ | ✓ | 0.8223 | 0.8149 | 0.8208 |
| ✓ | ✗ | ✗ | 0.8312 | 0.8176 | 0.8242 |
| ✓ | ✓ | ✗ | 0.8307 | 0.8168 | 0.8297 |
| ✓ | ✗ | ✓ | 0.8335 | 0.8235 | 0.8280 |
| ✓ | ✓ | ✓ | 0.8356 | 0.8277 | 0.8341 |

serve that the structure of the hypernetwork matters in such settings. Transformers may underfit facing smaller datasets, resulting in poor performances compared to the linear structure.

**Analysis on the distributions of hnet-prompt embeddings and hidden states.** We perform qualitative analyses on prompts and hidden states generated by HPrompt-CPT to investigate whether the hypernetwork can generalize domain information. As depicted in Fig. 5, We use t-sne map (van der Maaten and Hinton, 2008) to visualize the model output before and after training on all six domains in DAPset. For prompts, we observe that the generated prompt embeddings can effectively cluster similar domains together (e.g., overlapping embeddings for corpora $C_2$, $C_3$, and $C_5$ from the same paper dataset) while also achieving differentiation for dissimilar domains (e.g., distant embeddings for $C_1$ (restaurant) and $C_5$ (bio-chem)). This is an impressive result, i.e., it transfers the information across domains, making it easier for the LM to effectively adapt and generalize knowledge.

For hidden states, our model generates distinguishable hidden states for downstream task based on pre-trained domain information, i.e.,the initially mixed downstream representation ($D_1$ - $D_6$ in Fig. 5 top right) are successfully separated in Fig. 5 top left. For instance, the model assigns overlapping representations to similar tasks $D_2$ and $D_3$ (belonging to ACL and AI, respectively), while providing effective differentiation for unrelated tasks $D_1$ (restaurant) and $D_5$ (biology).

### 5.4 Ablation Study

Table 2 and 3 present the results of different designs of HPrompt-CPT on DAPset, where hyperparameters are fixed across all settings.

**Effectiveness of the main components.** To assess the impact of the hypernetwork, we replace

the hnet-prompt with progprompt (Razdaibiedina et al., 2023), which generates a new soft prompt for each domain and concatenates it and previously learned prompts while requiring domain-id during fine-tuning. As shown in Table 2 (rows 1 and 3), it results in a significant decrease in performances, particularly in adaptability, with an almost 1.77% decrease. It highlights the effectiveness of hnet-prompt in adapting and generalizing domain knowledge, providing great capacity for fine-tuning.

To examine the effect of the agreement and disagreement losses, we compare the results of training progressive prompt and hnet-prompt with and without them. It shows that incorporating the agreement and disagreement losses lead to a 1.15% and 1.20% improvement in $F\_Acc$ for the two models, respectively, demonstrating its efficiency in preventing CF. Furthermore, we observe that introducing the disagreement loss results in a 1.33% gain in $O\_Acc$, which is attributed to the incorporation of a wider range of domain knowledge for adaptation, as discussed in Sec. 4.2.

**Hypernetwork structure.** We further investigate the different designs of hypernetwork and present the results in Table 3 (top). First, we compare the network structure with the Linear layer or Multilayer Perceptron (MLP) (the top two rows), but they show poor adaptability and a higher level of CF. Interestingly, we find that the linear structure is more stable when facing a low-resource setting. Besides, we examine the performance of generating prompt embedding directly to show the significance of the component-based method introduced in Sec. 4.1. The results reveal that the component-based approach outperforms in generalization and preventing forgetting, benefiting from shifting the learning problem from remembering prompt to the weight vector which is a simple task.

**Agreement and disagreement loss objective.** We first replace the agreement loss with the conventional KD and the result are presented in the first row of Table 3 (middle). It shows agreement loss leads to a 1.06% improvement in adaptability while

Table 3: Ablation results on the hypernetwork structure and agreement/disagreement loss objective. Here, $\ell_a$ and $\ell_{da}$ denote the two losses. The content in parentheses represents the applied objective. The *logit* refers to minimizing the mean square error on logits. The *KL distance* aims to maximize the KL distance between hidden states. The *softmax variant* is to maximize the softmax on logits, following (Pagliardini et al., 2023).

| | | A_Acc | O_Acc | F_Acc |
|---|---|---|---|---|
| Hyper-network related | Linear network | 0.8332 | 0.8279 | 0.8331 |
| | MLP network | 0.8324 | 0.8210 | 0.8295 |
| | Generate prompt directly | 0.8336 | 0.8208 | 0.8305 |
| $\ell_a$ & $\ell_{da}$ related | $\ell_a$ (replaced with KD) | 0.8268 | 0.8230 | 0.8269 |
| | $\ell_a$ (logit) | 0.8310 | 0.8256 | 0.8295 |
| | $\ell_{da}$ (KL distance) | 0.8330 | 0.8253 | 0.8325 |
| | $\ell_{da}$ (softmax variant) | 0.8306 | 0.8242 | 0.8316 |
| | Ours | 0.8356 | 0.8277 | 0.8341 |

maintaining its ability to avoid forgetting, demonstrating its advantage in striking a balance of stability and plasticity for LM. Then, as it is unclear what kinds of objectives are most suitable to overcome forgetting, we test various objective functions for agreement and disagreement losses in Table 3 (middle). Ultimately, minimizing the KL-divergence of randomly prompted hidden states (agreement loss) and minimizing the orthogonal distance of current hidden states (disagreement loss) yield the best final performance of 83.41%.

# 6 Conclusion

This paper introduces HPrompt-CPT, a novel prompt-guided continual pre-training method towards anytime fine-tuning, which enables better performance when fine-tuned on seen and unseen domains. By training a hypernetwork to generate domain-specific prompts with agreement and disagreement losses, it results in (i) greater capacity on pre-trained domains by learning domain knowledge with generated prompts while preserving previous knowledge with random prompts, (ii) improved performance on unseen domains by retaining model plasticity with agreement loss and the ability of knowledge transfer with hypernetwork, and (iii) no need for domain-id during fine-tuning. We set a new SOTA on both well-established benchmark and a temporal shift benchmark.

# 7 Limitations

While we have evaluated our approach on two continual pre-training benchmarks, it remains unknown how well our method would perform on

benchmarks with severe domain conflicts. The domains in the benchmarks used in our paper are mostly transferable to each other. For example, the Domain "ACL" and "AI" in DAPset are highly related. We are not sure how will our method perform in a sequence of domains with little to no shared knowledge or even conflicts. In addition, we currently only test our method on the classification task, while the exploration of more types of downstream tasks is also important. Our future work will extend the benchmark to cover such cases.

Another problem for HPrompt-CPT is the selection of hypernetworks. Our experiments in Sec. 5.3 demonstrate that decreasing the size of the unlabeled corpus can cause the Transformer structure to underfit, while the Linear structure cannot capture all the information from a large corpus. In addition, we find the fine-tuning of hypernetwork is sensitive to the learning rate and weight decay. We aim to enhance the capacity and stability of our hypernetwork. Moreover, it is best to get a hypernetwork that can generalize well on downstream tasks without fine-tuning.

# Acknowledgements

The work was supported by grants from the National Key R&D Program of China (No. 2021ZD0111801) and the National Natural Science Foundation of China (No. 62022077).

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

# A    Additional related work

**Continual learning.** Continual Learning (CL) focuses on the problem of sequential learning from a stream of data that comes in different distributions. It is desired to extend the acquired knowledge to future tasks while avoiding catastrophic forgetting (CF) (McCloskey and Cohen, 1989) of the past tasks, and this has been successfully implemented in various fields, including computer vision (De Lange et al., 2021; Cha et al., 2021; Wang et al., 2022b), natural language processing (Sun et al., 2019; Huang et al., 2021; Qin and Joty, 2022), and Robotics (Wołczyk et al., 2021; Wolczyk et al., 2022). Traditional CL approaches fall into three types: regularization methods (Kirkpatrick et al., 2017; Li and Hoiem, 2017; Zenke et al., 2017; Aljundi et al., 2018), replay methods (Aljundi et al., 2018; Buzzega et al., 2020; Sun et al., 2022), and architecture-based methods (Serra et al., 2018; Li et al., 2019; Kang et al., 2022). However, designing CL methods for Language Modeling (LM) is different due to the two-stage training scheme (Devlin et al., 2019b; Qiu et al., 2020), where the model first learns universal language representations on a large corpus (pre-training stage), and then fine-tunes on downstream tasks (fine-tuning stage). Therefore, CL methods for LM can be categorized into Continual Pre-training and Continual Fine-tuning. In this paper, we focus on the continual pre-training.

**Continual Fine-tuning.** Continual Fine-tuning (FT) entails training a model on a stream of downstream tasks after its initialization. This approach requires the model to transfer knowledge to new tasks, avoid forgetting, and perform consistently well on all tasks learned before. Differing from the traditional CL, continual FT will mostly utilize the ability of a pre-trained language model. For instance, in replay methods, MbPA+ (de Masson D'Autume et al., 2019) uses BERT (Devlin et al., 2019a) to get the key representation of old examples for local adaptation when inference, while LAMOL (Sun et al., 2019) generates old examples through the generative ability of GPT2. IDBR (Huang et al., 2021) prevents forgetting by learning generic and task-specific knowledge using disentanglement-based losses. Recently, researchers have explored parameter-efficient tuning (Houlsby et al., 2019; Lester et al., 2021; Hu et al., 2022b) for CL. One kind of approaches (Ke et al., 2021b,a; Jin et al., 2021) involves using adapter modules (Houlsby et al., 2019) to incorporate task-specific parameters into frozen transformer layers, while the other (Qin and Joty, 2022; Zhu et al., 2022) uses soft prompts (Brown et al., 2020) to activate model ability to solve different tasks.

**Soft Prompt Learning.** Recent works (Wang et al., 2021a; Vu et al., 2022) have shown the potential of parameter-efficient learning in achieving multitask performance at low cost by stacking lightweight units. Their success attracts a surge of attention in adapting them to CL, especially for soft prompt tuning (Lester et al., 2021; Liu et al., 2022). These methods aim to transfer knowledge between tasks using prompts while avoiding forgetting. For example, LFPT5 (Qin and Joty, 2022) employs a large soft prompt that is continually trained on all tasks while also distilling previous knowledge. Continual Prompt Tuning (Zhu et al., 2022) uses initialization, memory replay, and AGEM (Chaudhry et al., 2019) technologies to facilitate prompt forward/backward transfer. ProgPrompt (Razdaibiedina et al., 2023) leverages previously learned prompts by concate-

nating them with new embeddings. While most of these methods require task-id to select the appropriate prompt, DualPrompt (Wang et al., 2022a) and L2P (Wang et al., 2022c) create a prompt pool and learn a cluster-based mapping from input data to a specific prompt. Furthermore, CodaPrompt (Smith et al., 2023) suggests learning a composition of the prompt pool that replaces index operations with a backpropagation-based approach.

**Konwledge distillation.** Knowledge distillation (KD) (Hinton et al., 2015) is a widely-used technique for improving performance and efficiency in various tasks, such as model compression (Meng et al., 2021; Chen et al., 2022) and transfer learning (Xu et al., 2020; Fang et al., 2021). KD has also been applied in continual learning to transfer knowledge learned from old tasks to new ones and thus prevent forgetting (Chuang et al., 2020; Dong et al., 2021; Ke et al., 2021a). However, previous approaches mainly focused on aligning the entire feature space, which can limit the adaptation ability of the model. Instead, we propose using an agree and disagree loss to decompose KD into prompt and feature space, achieving a better balance between plasticity and stability.

## B  Dataset Details

**DAPset.** It is a benchmark for continual domain adaptive pre-training, constructed by (Ke et al., 2023). It consists of six domains, each with an unlabeled corpus and a corresponding downstream task classification dataset. They are from two large datasets, while 3 of them are about reviews: Yelp Restaurant (Xu et al., 2019)/ Restaurant (Ding et al., 2008), Amazon Phone (Ni et al., 2019)/ Phone (Ding et al., 2008), Amazon Camera (Ni et al., 2019)/ Camera (Ding et al., 2008) and 3 of them are academic papers: ACL papers (Lo et al., 2020)/ ACL-ARC (Jurgens et al., 2018), AI papers (Lo et al., 2020)/ SCIERC (Luan et al., 2018), and PubMeb papers (Lo et al., 2020)/ CHEMPROT (Kringelum et al., 2016). The front one is the unlabeled corpus and the latter one is the corresponding downstream task. We show the statistics of these datasets in Table 4.

The downstream tasks in DAPset can be divided into the following 4 types. (1) Aspect sentiment classification: given an aspect (e.g., *environment* in a restaurant review) and the corresponding review text, classify it into positive, negative, or neutral sentiment. (2) Citation intent classification: given

a sentence contains a citation, classify the intent of this citation. (3) Relation classification: given a sentence together with its entities, classify the relation of these entities. (4) Chemical-protein interaction classification: given a sentence containing a pair of chemicals and proteins, classify the interaction between these two.

**TWEET.** We develop a new benchmark, TWEET, following (Jin et al., 2022), to simulate the distribution shift over time. The dataset is collected by the Archive team [2] and we select the text data from 2015 to 2019, dividing it into five time periods and creating five domain corpora. The text data was pre-processed the text data according to (Nguyen et al., 2020) with the hashtags removed. We randomly select 300 MB of data from each period, and the statistics of each domain is present in Table 4.

For the downstream task, we build a single-label hashtag prediction task for each domain following Gong and Zhang (2016). We count the 10 most frequently hashtags (e.g., "#hiring", "#music") in each time period and extract 700 tweet texts for each label, including 200 tweets for training, 200 tweets for validation, and 300 tweets for testing. Before the text is input into the model, we remove the hashtags themselves and ask the model to predict the most appropriate hashtag for the current sentence.

## C  Implementation Details

We adopt Roberta-BASE (Liu et al., 2019) as our backbone language model and 6-layer Transformer (Vaswani et al., 2017) as our hypernetwork. In the pre-training phase, we apply a masked language model head after the LM, which is then replaced with a classification head during fine-tuning. Each downstream task has its own classification head. In pre-training, the trainable parameters depend on the algorithm design, while in fine-tuning, all parameters, including the language model and added model, are trainable.

The maximum input sequence length is set to 164, following (Ke et al., 2023), and we use an Adam optimizer with a weight decay of 0.01 for both pre-training and fine-tuning. During pre-training, the learning rate is set to 1e-4 and batch size to 128. We train for 5K and 2.5K steps for each domain in DAPset and Tweet, respectively, which is roughly a full pass through the domain

---

[2]https://archive.org/details/twitterstream

Table 4: Statistics of datasets for DAPset and TWEET.

| Benchmarks | | Unlabeled Corpus | | | Downstream Task Datasets | | | |
| --- | --- | --- | --- | --- | --- | --- | --- | --- |
| | Source | Dataset | Size | Dataset | Classification Task | #Training | #Testing | #Classes |
| DAPset | Reviews | Yelp Restaurant | 758MB | Restaurant | Aspect Sentiment | 3,452 | 1,120 | 3 |
| | | Amazon Phone | 724MB | Phone | Aspect Sentiment | 239 | 553 | 2 |
| | | Amazon Camera | 319MB | Camera | Aspect Sentiment | 230 | 626 | 2 |
| | Academic Papers | ACL Papers | 867MB | ACL-ARC | Citation Intent | 1,520 | 421 | 6 |
| | | AI Papers | 507MB | SCIERC | Relation | 2,260 | 2,388 | 7 |
| | | PubMed Papers | 989MB | CHEMPROT | Chemical-protein Interaction | 2,667 | 7,398 | 13 |
| TWEET | Tweet | $Tweet\_i$ | 300MB | $Hashtag\_i$ | Hashtag Prediction | 2,000 | 3,000 | 10 |

Table 5: Performance on different order of domains on DASSET.

| Domain Order | Derpp | | | DAS | | | Ours | | |
| --- | --- | --- | --- | --- | --- | --- | --- | --- | --- |
| | $A\_Acc$ | $O\_Acc$ | $F\_Acc$ | $A\_Acc$ | $O\_Acc$ | $F\_Acc$ | $A\_Acc$ | $O\_Acc$ | $F\_Acc$ |
| REST:ACL:AI:PHONE:PUBMED:CAMERA | 0.8245 | 0.8174 | 0.8239 | 0.8261 | 0.8164 | 0.8251 | **0.8356** | **0.8277** | **0.8341** |
| REST:PHONE:PUBMED:CAMERA:AI:ACL | 0.8262 | 0.7815 | 0.8180 | 0.8241 | 0.7830 | 0.8148 | **0.8351** | **0.7939** | **0.8244** |
| CAMERA:PUBMED:PHONE:AI:ACL:REST | 0.8263 | 0.8023 | 0.8166 | 0.8273 | 0.8032 | 0.8127 | **0.8317** | **0.8099** | **0.8264** |
| PHONE:ACL:PUBMED:REST:CAMERA:AI | 0.8175 | 0.8319 | 0.8205 | 0.8193 | 0.8260 | 0.8155 | **0.8269** | **0.8366** | **0.8278** |

data. We set the prompt length to 50, the size of prompt components to 100, and the size of memory buffer to 300. As for the trade-off hyperparameters, we set $\lambda_r$ to 1, $\lambda_a$ to 0.01, and $\lambda_{da}$ to 0.01. During fine-tuning, the learning rate is set to 3e-5 and batch size to 16. We train on the downstream datasets for 15 epochs with an early stopping mechanism. Unless otherwise stated, the same hyper-parameters are used in all experiments.

## D  Robustness on different orders

As several works (Yoon et al., 2020; Evron et al., 2022) suggest that the task order may significantly affect the performance of CL approaches, we also conduct experiments to test our robustness to different task orders. We run the methods on several orders of the DAPset benchmark and list results in Table 5. Our method shows an average improvement of 0.99%, 1.22%, and 1.36% on the three metrics compared to DAS, demonstrating its robustness.