# OpenReview forum: "Towards Anytime Fine-tuning: Continually Pre-trained Language Models with Hypernetwork Prompts"
_EMNLP/2023/Conference — EMNLP 2023 Findings_

### Official Review · Reviewer_JHzV · 2023-07-31

**Typos Grammar Style And Presentation Improvements:** 1. Figure 1, it's recommended to give…
**Soundness:** 3

**Excitement:**

2: Mediocre: This paper makes marginal contributions (vs non-contemporaneous work), so I would rather not see it in the conference.

**Paper Topic And Main Contributions:**

This paper proposes a prompt-guided continual pre-training method (HPrompt-CPT). It involves a network to generate prompts by proposed agreement and disagreement losses. The design takes into account: adaptability, generalizability and forgetting. Experiments show that it outperforms other fine-tuning (FT) method on all 3 aforementioned aspects

**Questions For The Authors:**

1. How are the prompt components generated?
2. Line 360: How random prompt can simulate the condition to active generic or learned domain knowledge?
3. What is the rationale to use KL-Divergence instead of other metrics?
4. Formula 3: to minimize KL-Divergence, why does it define loss as the opposite value of KL-Divergence?

**Reasons To Accept:**

1. This paper takes into account 3 key capabilities of continual pre-trained models: adaptability, generalizability and forgetting.
2. The experiments covers a wide range of existing fine-tuning methods

**Reasons To Reject:**

1. The paper claims to propose an evaluation protocol named anytime fine-tuning. However, this is not something new. It's a common practice to test on seen, seeing and unseen datasets;
2. This paper introduces Hnet-prompt module (including a 6-layer Transformer) to generate prompts and run both current and previous language models (LM) multiple times to get the losses which largely increases the computation costs;
3. This method are only tested with Roberta. Its generalizability on other LM is unclear;
4. This method is only targeting prompt learning based light-weight fine-tuning which limits its usage with other light-weight fine-tuning methods;
5. Experiments were done on only 2 datasets including one own-built dataset, which hinders the representativity of experiment results.

**Reproducibility:**

3: Could reproduce the results with some difficulty. The settings of parameters are underspecified or subjectively determined; the training/evaluation data are not widely available.

**Reviewer Confidence:**

2: Willing to defend my evaluation, but it is fairly likely that I missed some details, didn't understand some central points, or can't be sure about the novelty of the work.

---

> ### Author Rebuttal · Authors · 2023-08-29
>
> We sincerely appreciate your comments on this paper. You may find our response below for your major and minor concerns. We would appreciate it if you could let us know if you have any further concerns.
>
> ##### Q1: Relationship between anytime fine-tuning and the common practice to test on seen, seeing, and unseen datasets
>
> > We would like to humbly clarify that the proposed **anytime fine-tuning for evaluating continual pre-training** is not equivalent to **the common practice of testing on seen/seeing/unseen datasets under continual fine-tuning [5, 6]**.
> >
> > - First, continual pre-training and continual fine-tuning fall into two distinct categories of continual learning scenarios in language models, whose differences are summarized in the following table.
> >
> >   |                        | Training                               | Evaluation                                                 | Goal                                   |
> >   | ---------------------- | -------------------------------------- | ---------------------------------------------------------- | -------------------------------------- |
> >   | Continual fine-tuning  | Streaming **downstream labeled** tasks | Direct test accuracy on seen/seeing/unseen tasks           | One model serving all downstream tasks |
> >   | Continual pre-training | Streaming **unlabeled** corpora        | Test accuracy **after fine-tuning** on domain-related tasks | Well-generalized initialization        |
> > - Though it is common in continual fine-tuning to evaluate unseen tasks, the current evaluation for continual pre-training assumes the greater generalization ability of an LM via continual updates and focuses on fine-tuning in seen/seeing datasets. Our empirical results in Figure 2, however, signify the unexpected generalization performance drop of a continually pre-trained LM in unseen tasks. **This empirical finding, to our best knowledge, is novel and motivates our first proposal of evaluating a continually pre-trained model in unseen tasks**.
>
> ##### Q2: The hypernetwork and generation of previous LM increases time complexity
>
> > - We present a table in response to Reviewer 2's Question 2, comparing the **training cost** and **performance** of several algorithms. Our training complexity is **comparable to das** [3], the previous sota method for continual pretraining proposed in ICLR 2023, while we achieve a significant improvement.
> > - Additionally, we have conducted experiments with a simpler hypernetwork structure using a linear layer. This modification decreases the time cost while maintaining similar performance, providing **flexibility in selecting** the network according to different training requirements.
> > - For the cost from running current and previous LMs multiple times, we can adopt a **preprocessing** approach to calculate and save the historical representations of the samples before the forward pass. Furthermore, our agreement and disagreement loss can operate independently. In time-sensitive scenarios, we can customize a suitable algorithm by **removing specific losses**.
>
> ##### Q3: Its generalizability on other LM is unclear. And experiments were done on only 2 datasets.
>
> > - We extended the algorithm to the **GPT2 model** and ran on **two continual learning generation task datasets**, namely DecaNLP and Reviews. For detailed experimental settings and results, please refer to our response to Reviewer 2's Question 1. Our method achieves **state-of-the-art** performance and exhibits outstanding generalization ability.
>
> ##### Q4: The design of prompt learning limits its usage with other light-weight fine-tuning methods.
>
> > - Prompt learning, which involves concatenating prompt embeddings with input data, can be considered as an augmentation to the data. Consequently, other lightweight fine-tuning methods can be applied on top of the prompted input without any negative effects.
> >
> > - For instance, during the fine-tuning phase, we can add adapter layers in the language model while keeping the hypernetwork and backbone model frozen.  These adapter layers will acquire task-specific knowledge with the assistance of a pre-trained hypernetwork that transfers domain knowledge through the prompt. The fixed hypernetwork is sufficiently generalized for downstream tasks, achieving a final accuracy of 0.8326 (0.8341 for full fine-tuning).
>
> ##### Q5: How are the prompt components generated?
>
> > - Prompt components, **analogous to a set of basis vectors**, are a set of prompt embeddings that are randomly initialized, trainable and optimized through gradient descent. The well-trained prompt components are supposed to offer **greater generalization to future domains** as long as the prompt components are as mutually exclusive as possible. For example, a prompt embedding directly optimized for the domain of "ACL papers" does not directly apply to the domain of "AI papers" due to the domain difference; however, one of the prompt components learned on "ACL papers", e.g., "deep learning", can be combined with another component of "statistics" to generalize to the domain of "AI papers".
>
>
> ##### Q6: How random prompt can simulate the condition to active generic or learned domain knowledge?
>
> > - The oracle generic and previous knowledge lies in the pre-training data, which usually remains inaccessible. We seek an approach that simulates the generic knowledge, which is expected to be **irrelevant to the current domain**. Since randomly chosen prompts hardly have the chance to share the knowledge with the current domain, they can be used to simulate the generic and previous knowledge. This idea has also been leveraged in previous works [7].
>
> ##### Q7: What is the rationale to use KL-Divergence instead of other metrics?  And why use the opposite value of KL-Divergenc?
>
> > - KL divergence distance is a common choice for knowledge distillation and was first introduced by Hinton [1]. It has been widely used in the field of continual learning [2, 3, 4]. Compared to loss functions like MSE, minimizing KL divergence is more sensitive to small differences and has a more stable gradient. In our work, we treat feature vectors as discrete representations of probability distributions and use the constraint of KL divergence to prevent the model forgetting, following [3, 4].
> >
> > - The minus sign in Formula 3 is a typo. Thanks for pointing it out.
>
> ##### Q8: There are many small mistakes.
>
> > - Thanks for pointing out the above-mentioned mistakes and typos in our paper. We will carefully polish our paper again following your great feedback.
>
> [1] Geoffrey Hinton, et al. "Continual Pre-training of Language Models" Arxiv 2015.
>
> [2] Xisen Jin, et al. "Lifelong Pretraining: Continually Adapting Language Models to Emerging Corpora" NAACL 2022.
>
> [3] Zixuan Ke, et al. "Continual Pre-training of Language Models" ICLR 2023.
>
> [4] Wuyang Chen, et al. "Lifelong Language Pretraining with Distribution-Specialized Experts" ICML 2023.
>
> [5] Xisen Jin, et al. "Learn Continually, Generalize Rapidly: Lifelong Knowledge Accumulation for Few-shot Learning" EMNLP 2021.
>
> [6] Thomas Scialom, et al. "Fine-tuned Language Models are Continual Learners" EMNLP 2022.
>
> [7] Kevin Meng, et al. "Mass-Editing Memory in a Transformer" ICLR 2023.

---

### Official Review · Reviewer_Q6RZ · 2023-08-04

**Soundness:** 4

**Excitement:**

3: Ambivalent: It has merits (e.g., it reports state-of-the-art results, the idea is nice), but there are key weaknesses (e.g., it describes incremental work), and it can significantly benefit from another round of revision. However, I won't object to accepting it if my co-reviewers champion it.

**Paper Topic And Main Contributions:**

The authors propose a novel approach for continual pre-training of language models and introduce an evaluation protocol. Unlike the previous continual pre-training methods that mainly focus on the performance of past and current domains, the proposed evaluation protocol, called anytime fine-tuning, evaluates the model performance not only on past and current domains but also on unseen future domains for LM's generalization ability. The authors utilize hyper-network for prompt generation and introduce agreement and disagreement losses for adaptability and generalization, respectively. The proposed method is compared against standard CL methods and recent continual domain adaptation methods, and shows strong performance in the anytime fine-tuning protocol.

**Reasons To Accept:**

1. The paper is well written.
2. The paper suggests an important evaluation protocol that has been overlooked in previous literature
3. The proposed method shows strong performance.
4. The advantage of prompt generation via hyper-network is reasonable and intuitive.

**Reasons To Reject:**

1. It's not obvious how the proposed method prevents forgetting.
2. The design of the experiments for downstream task evaluation is not clear. For instance, it's not specified whether all prompts are fine-tuned for downstream tasks or if the domain of the downstream task is known beforehand.
3. Code is not provided

**Reproducibility:**

2: Would be hard pressed to reproduce the results. The contribution depends on data that are simply not available outside the author's institution or consortium; not enough details are provided.

**Reviewer Confidence:**

3: Pretty sure, but there's a chance I missed something. Although I have a good feel for this area in general, I did not carefully check the paper's details, e.g., the math, experimental design, or novelty.

---

> ### Author Rebuttal · Authors · 2023-08-29
>
> We sincerely appreciate your constructive comments to improve our paper. We detail our response below point by point. Please kindly let us know if our response addresses the questions you had for this paper.
>
> ##### Q1: It's not obvious how the proposed method prevents forgetting.
>
> > - In our method, we employ three mechanisms to address forgetting, as follows:
> >
> >   - Hypernetwork prompt: It **learns domain-specific knowledge through the hypernetwork** and minimizes updates in the backbone model, thereby reducing the overwrite of previously learned knowledge. It mitigates forgetting ($A\_Acc$ - $F\_Acc$) from 1.00% to 0.70%.
> >
> >   - Disagreement loss: It **promotes the exclusiveness** of generated hidden states for the current domain, minimizing interference to established knowledge in the LM. It mitigates forgetting from 0.70% to 0.55%.
> >
> >   - Agreement loss: It **aligns the predictions** of previous and current language models prompted by random prompts to preserve the generic and learned domain knowledge. It mitigates forgetting from 0.55% to 0.14%.
>
>
> ##### Q2: The design of the experiments for downstream task evaluation is not clear.
>
> > - In the anytime fine-tuning setting, the downstream task evaluation phase **does not involve providing domain ID** for fine-tuning and testing. This is more in line with the real-world scenarios, where it is difficult to determine the domain of each incoming downstream task, especially for unseen domains.
> >
> > - In our HPrompt-CPT algorithm, during the downstream task evaluation phase, we utilize the trained **hypernetwork** to map the samples to prompt embeddings, which generalizes the pre-trained domain knowledge to downstream task **without the need of domain-id**. The visualized distribution in Fig.5 verifies the alignment of the generated prompts for unlabeled corpus and downstream task within a specific domain. Moreover, the hypernetwork undergoes updates during the fine-tuning phase to enhance downstream adaptation.
> >
> > - We explain this process in line 325 of the paper and will refine the text in subsequent revisions to improve clarity.
>
> ##### Q3: Code is not provided
>
> > - We guarantee that the code will be made available to the readers. In the camera-ready version of the paper, we will include a link to the code repository.

---

### Official Review · Reviewer_YTnf · 2023-08-09

**Soundness:** 4

**Excitement:**

3: Ambivalent: It has merits (e.g., it reports state-of-the-art results, the idea is nice), but there are key weaknesses (e.g., it describes incremental work), and it can significantly benefit from another round of revision. However, I won't object to accepting it if my co-reviewers champion it.

**Paper Topic And Main Contributions:**

This paper proposes a continual pre-training method for language models called HPrompt-CPT. It uses a hypernetwork to generate domain-specific prompts during continual pre-training. This approach aims to balance adaptation, generalization, and mitigate forgetting for anytime fine-tuning effectiveness.
The main contributions are:

They propose a anytime fine-tuning evaluation protocol for continual pre-training and show that existing methods struggle with adaptability and generalization.

They propose the HPrompt-CPT approach with a hypernetwork prompt module and agreement/disagreement losses. The hypernetwork generates prompts that capture domain knowledge while generalizing across domains. The losses aim to preserve generalization while mitigating forgetting.

They perform experiments on two datasets and show that HPrompt-CPT achieves better adaptation, generalization and less forgetting, outperforming baselines.

**Reasons To Accept:**

· The proposed anytime fine-tuning evaluation protocol provides a more comprehensive view of continual pre-training models.
· The hypernetwork prompt module and agreement/disagreement losses are novel approaches that aim to balance the three important aspects of continual pre-training.
· The experiments demonstrate the effectiveness of the proposed HPrompt-CPT approach.

**Reasons To Reject:**

· The experiments are performed on only two datasets, more benchmarks would strengthen the claims.
· The hypernetwork prompt module adds complexity, simpler approaches could be explored first.
· The empirical analysis of existing methods is limited, a more thorough comparison is needed.
· The approach is focused on continual pre-training, its effectiveness on other continual learning scenarios remains unclear.

**Reproducibility:**

3: Could reproduce the results with some difficulty. The settings of parameters are underspecified or subjectively determined; the training/evaluation data are not widely available.

**Reviewer Confidence:**

3: Pretty sure, but there's a chance I missed something. Although I have a good feel for this area in general, I did not carefully check the paper's details, e.g., the math, experimental design, or novelty.

---

> ### Author Rebuttal · Authors · 2023-08-29
>
> We sincerely appreciate your constructive comments to improve our paper. We detail our response below point by point. Please kindly let us know if our response addresses the questions you had for this paper.
>
> ##### Q1: The experiments on more benchmarks would strengthen the claims.
>
> > - To further demonstrate the advancement of HPrompt-CPT, we extended this algorithm to **the GPT2 model** and conducted experiments on **two continual learning generation task datasets**, DecaNLP and Reviews, used in [1, 2].
> >
> > - The **brief information** for these two datasets is as follows: 1) DecaNLP consists of four domains: Question answering, Semantic Parsing, Sentiment role labeling, and Zero-shot Relation Extraction. The reviews dataset includes five domains: Yahoo, Yelp, Amazon, AGnews, and DBPedia. 2) For pre-training, we remove the label for most of the data and use it as a domain corpus. For downstream task evaluation, we select the labeled samples from the remaining data as the corresponding downstream dataset. 3) The unlabeled corpus size: DecaNLP ≤ 8M tokens, Reviews ≤ 16M tokens. The downstream task size: DecaNLP = 1500 samples, Reviews = 250 samples.
> >
> > - We compared our methods NCL, EWC, and DERPP, and the results are as follows:
> >
> >    |       | DecaNLP |         |         |  Reviews |         |         |
> >    |-------|---------|---------|---------|---------|---------|---------|
> >    |       | A_Acc   | O_Acc   | F_Acc   | A_Acc   | O_Acc   | F_Acc   |
> >    | NCL   | 57.92 | 66.93 | 59.23 | 60.66 | 78.75 | 59.56 |
> >    | Ewc   | 56.62 | 66.09 | 56.40 | 59.05   | 78.36 | 58.97  |
> >    | Derpp | 56.85 | 65.78 | 57.09 | 61.36 | 79.35 | 61.93 |
> >    | Ours  | **61.83** | **69.61** | **61.88** | **63.66** | **82.03** | **63.76** |
> >
> >   - Our algorithm achieved **state-of-the-art** performance on both datasets, and its performance in terms of **generalization** is remarkable. The performance drop of Derpp on the first dataset can be attributed to the **sample imbalance issue**, which resulted in the buffer stored by Derpp being unrepresentative and thus leading to subpar results.
>
> ##### Q2: The hypernetwork prompt module adds complexity.
>
> > - We present a table below comparing the **training cost** and **performance** of several algorithms:
> >
> >    |           | DAPset  |        |        |                         |
> >    |-----------|---------|--------|--------|-------------------------|
> >    |           | A_Acc   | O_Acc  | F_Acc  | Time cost [sec / batch] |
> >    | NCL       | 0.8298  | 0.8189 | 0.8198 | **0.217**                   |
> >    | Derpp     | 0.8245  | 0.8174 | 0.8239 | 0.430                   |
> >    | DAS       | 0.8221  | 0.8164 | 0.8251 | 0.685                   |
> >    | Ours(Lin) | 0.8332  | **0.8279** | 0.8331 | 0.601                   |
> >    | Ours      | **0.8356**  | 0.8277 | **0.8341** | 0.730                   |
> >
> > - The results indicate that our method indeed introduces more time overhead compared to simple algorithms like NCL and Derpp.  However, the time overhead is **comparable to das** [3], a method proposed in ICLR 2023, while we achieve **significant improvement** in performance.
> > - We also experimented with a simpler structure for the hypernetwork by using a linear layer. This approach reduced the time cost while maintaining similar performance, providing **flexibility in choosing the network** based on different training requirements.
>
>
>
> ##### Q3: The empirical analysis of existing methods is limited.
>
> > - **Naive continual learning serves as a strong baseline for continual pre-training,** which outperforms EWC, LWF, and Coda-Prompt. This may be because 1) traditional continual learning methods **struggle** with large amounts of task data and self-supervised objectives. 2) Self-supervised tasks are naturally well-suited for continual learning as they provide a **smoother loss path**.
> >
> > - **Baselines excel at preventing forgetting but struggle with adaptability.** The forgetting metrics, $A\_Acc - F\_ACC$, for the baselines are mostly negative, indicating that forgetting is not the main reason for the low final accuracy. In contrast, the adaptation to the learned domain is the **key weakness**. Parameter-efficient tuning methods such as One-Adapter, One-Prompt, and CoDA-Prompt result in low $A\_Acc$, as it is difficult for the small trainable portion of the model to incorporate a significant amount of domain knowledge. Similarly, methods that include regularization terms on trainable model size like EWC and DAS also face challenges in learning domain knowledge. Otherwise, Derpp and LwF show relatively better adaptability, as they both employ knowledge distillation, suggesting it could be a better choice.
> >
> > - **Most methods sacrifice generalization to unseen domains**. In the DAPset, the majority of baselines exhibit lower $O\_Acc$ compared to the initial roberta. EWC, which primarily prioritizes the importance of learned knowledge, shows a significant drop in performance, while LwF protects the output of the initial model and achieves the best generalization among the baselines. Replay-based methods also result in low generalization due to their tendency to overfit on the memory examples.
>
> ##### Q4: Its effectiveness on other continual learning scenarios remains unclear.
>
> > - **Continual pre-training** entails learning from streaming unlabeled corpora during the pre-training phase, with the purpose of enhancing the generalization capabilities for all coming downstream tasks.
> >
> > - It **differs greatly** from continual fine-tuning, and we have summarized their **characteristics in the table** provided in response to Reviewer 4's Question 1. These two scenarios represent distinct lines of research within the community. Researchers [1, 2, 3] have designed methods specifically tailored to each scenario and **do not explore the effectiveness** of these methods in both scenarios.
> >
> > - Our method focus on an interesting problem called **anytime fine-tuning** in continual pre-training, which holds significant value for real-world development. And we provide comprehensive analysis of our method in continual pre-training to demonstrate its advancement.
>
>
>
> [1] Fan-Keng Sun, et al. “LAMOL: LAnguage MOdeling for Lifelong Language Learning” ICLR 2020.
>
> [2] Yingxiu Zhao, et al. “Semi-Supervised Lifelong Language Learning” EMNLP 2022.
>
> [3] Zixuan Ke, et al. "Continual Pre-training of Language Models" ICLR 2023.

---

### Official Review · Reviewer_ke4a · 2023-08-21

**Soundness:** 3

**Excitement:**

4: Strong: This paper deepens the understanding of some phenomenon or lowers the barriers to an existing research direction.

**Paper Topic And Main Contributions:**

The main problem that the paper addresses is existing continual pretraining methods do not perform well in terms of generalization and adaptability, while they work well against catastrophic forgetting. The main contributions of the paper are
- design an architecture to use a hypernetwork to generate domain specific prompts to guide the continual pretraining.
- introduced agreement and disagreement losses to aid the training, and conducted ablation studies for these components.
- the designed architecture achieved better results in terms of generalization, adaptability, and can do well against forgetting, comparing to existing SOTAs.
- conducted a large scale experiment to cover 7 methods of continual pretraining on two datasets

**Questions For The Authors:**

- What would be the next sets of experiments to conduct, given the current ones are focused on classification tasks?
- Can you expand on the significance on the t-SNE plot? Since the model is fine tuned, it seems natural that the embeddings and hidden states would have changed. Please explain in more details what the domains and downstream tasks are with more examples.
- The question of order of continual pretraining has always been on my mind, so more explanation would be helpful.
- More explanation on prompt components and how to choose the hyperparameters would be supportive to the experiment details.
- Although prompts are pretrained with the hypernetwork, can you provide more justification of why this is interpreted as p(output | input, domain)?
- In ablation studies and the rest of the paper, there is a lack of confidence interval and the quantitative values are quite close.

**Reasons To Accept:**

- The motivation of the paper is very useful, since pushing the boundary of downstream tasks adaptability, generalization, and avoiding catastrophic forgetting is crucial to real life application and research
- There is a fairly comprehensive experiment done against the several SOTAs in the realm of continual pretraining done.
- The empirical results show promising improvement upon the weakness of current methods.

**Reasons To Reject:**

- As acknowledged by the authors, the downstream application is currently limited to classification. The experiment although is comprehensive in terms of approaches, on the dataset it is not too diverse.
- The performance of the experiment in the third domain seems to be also dropping like the other methods, so it's not too strongly supporting the argument in the perspective of generalization. More studies and focus can be put on this.

**Reproducibility:**

4: Could mostly reproduce the results, but there may be some variation because of sample variance or minor variations in their interpretation of the protocol or method.

**Reviewer Confidence:**

3: Pretty sure, but there's a chance I missed something. Although I have a good feel for this area in general, I did not carefully check the paper's details, e.g., the math, experimental design, or novelty.

---

> ### Author Rebuttal · Authors · 2023-08-29
>
> We sincerely appreciate your constructive comments to improve our paper. We detail our response below point by point. Please kindly let us know if our response addresses the questions you had for this paper.
>
> ##### Q1: The downstream application is currently limited to classification.  What would be the next sets?
>
> > - For continual pre-training, the type of downstream tasks is **not restricted** by the algorithm. Our proposed method, HPrompt-CPT, is also **applicable to generation** downstream applications. However, due to the limitations of the backbone model we used (Roberta) and the datasets collected, we did not test the algorithm's performance on generation tasks in the paper.
> >
> > - To further demonstrate the advancement of HPrompt-CPT, we extended the backbone model to the GPT2 and **ran on two continual learning generation task datasets**, namely DecaNLP and Reviews. For detailed experimental settings and results, please refer to our response to Reviewer 2's Question 1. Our method achieves **state-of-the-art performance** and exhibits outstanding generalization ability.
>
>
>
> ##### Q2: The performance of the experiment in the third domain seems to be also dropping like the other methods
>
> > - Hprompt-CPT prevents and improves the model generalization to unseen domains through three mechanisms:
> >   - Using **hypernetwork** to map input to embedding, allowing the model to **generalize past sample knowledge** to unknown samples.
> >   - **Protect the general knowledge** in language model through Agreement loss.
> >   - **Increasing the hypothesis space** of the language model output through Disagreement loss, enhancing generalization.
> >
> > - When learning domain 0, our **mechanisms face challenges** in preventing performance degradation on the conflicting domain 3. In this scenario, the first and third mechanisms are unable to generalize the knowledge to a conflicting future tasks, and the protection provided by the second mechanism is insufficient to counteract the shifts caused by adapting to the conflicting task.
> >
> > - But in the **subsequent process**, NCL and DAS algorithms continue to experience performance degradation on domain 3 since they lack designs specifically aimed at preserving generalization. In contrast, our algorithm, through the application of the three mechanisms, can gradually generalize the previous learned knowledge to new tasks and decrease the shift in language model, thereby **avoiding further performance decline**.
> >
> > - In summary, our capacity to generalize on unseen domains strengthens as we learn more domains, thereby achieving the objective of anytime fine-tuning.
>
>
> ##### Q3: Can you expand on the significance on the t-SNE plot?
>
> > The t-SNE plot is used to demonstrate that HPrompt-CPT achieves the learning and preservation of domain knowledge during continual pretraining, which also well transferred to downstream tasks without relying on domain IDs.
> >
> > - For prompts, the hypernetwork is expected to learn the correlation between domains and allocate corresponding prompts. The bottom right corner in Fig.5 illustrates this point. Our hypernetwork **assigns similar embeddings to similar domains** (e.g., overlapping embeddings for corpora C2, C3, and C5 from the same paper dataset) while also achieving **differentiation for dissimilar domains** (e.g., distant embeddings for C1 (restaurant) and C5 (bio-chem)). Furthermore, the hypernetwork is expected to identify the domain to which the coming downstream task belongs and allocate the corresponding embeddings. The results in Fig.5 show that the **prompt distributions generated for D2, D3, and D5 have relative positions consistent with C2, C3, and C5**, validating our hypothesis.
> >
> > - For hidden states, effective models can generate distinguishable hidden states for downstream task based on pre-trained domain information. The results in the top right corner of Fig.5 indicate that our model successfully possesses this capability. For instance, the model assigns **overlapping representations to similar tasks** D2 and D3 (belonging to ACL and AI, respectively), while providing **effective differentiation for unrelated tasks** D1 (restaurant) and D5 (biology).
>
>
>
> ##### Q4: More explanation to the question of order of continual pretraining.
>
> > We have illustrated in Appendix D and Table 5 the results of the order of continual pre-training. We conclude that
> >
> > - **among all orders**, the order with highly similar adjacent tasks leads to smaller changes in model parameters, thereby contributing a flatter loss surface and greater performance. This aligns with the works of [3, 4].
> >
> > - **among all methods**, Derpp and the proposed HPrompt-CPT demonstrate greater robustness against various orders. Though Derpp requires experience replay to promote similarity between adjacent tasks, HPrompt-CPT minimizes changes in the backbone and improves robustness via the hypernetwork-based and domain-specific prompts.
>
>
> ##### Q5: More explanation on prompt components?
>
> > - Prompt components, **analogous to a set of basis vectors**, are a set of prompt embeddings that are randomly initialized, trainable and optimized through gradient descent.
> >
> > - Compared to learning a prompt embedding for each domain, introducing prompt components and **instead learning the weights that combine these components** offer the following two benefits:
> >
> >   - **greater generalization to future domains** as long as the prompt components are as mutually exclusive as possible. For example, a prompt embedding directly optimized for the domain of "ACL papers" does not directly apply to the domain of "AI papers" due to the domain difference; however, one of the prompt components learned on "ACL papers", e.g., "deep learning", can be combined with another component of "statistics" to generalize to the domain of "AI papers".
> >
> >   - **less forgetting** as the number of parameters to learn for each domain reduces from the full size of a prompt embedding to the number of prompt components.
> >
> > - Empirically, we **validate the above benefits in Table 3**, where adapting prompt components yields better results compared to directly generating prompts through linear projection.
>
>
> ##### Q6: How to choose hyperparameters?
>
> > Hyperparameters at different parts of the algorithm have different impacts on the adaptability, generalization, and forgetting of modes. We list the choices of several important hyperparameters below.
> >
> > - **Learning rate**: It plays a crucial role in the fine-tuning stage, where a small $lr=1e-5$ failed to fit the task for unseen domains, leading to an $O\_Acc$ of 0.8197, compared to $lr=3e-5$, which achieved 0.8297. Conversely, a large $lr=1e-3$ resulted in reduced performance in learned domains, resulting in an $F\_Acc$ of 0.8216 compared to 0.8306 for $lr=3e-5$.
> >
> > - **Prompt token length**: We find that a larger prompt length (80) may impede generalization for downstream tasks, resulting in a drop in $A\_Acc$ from 0.8356 to 0.8232.
> >
> > - **Weight for agreement and disagreement losses**: A higher agreement loss demonstrates improved performance in mitigating forgetting (0.0101 -> 0.0030), but it leads to poorer generalization (0.8274 -> 0.8198) due to alignment. On the other hand, a higher disagreement loss shows better performance in alleviating forgetting (0.0032 -> -0.0016) and improves generalization (0.8245 -> 0.8289), but it results in poorer adaptability (0.8325 -> 0.8238) due to exclusiveness.
>
>
>
> ##### Q7: Why the prompted language model is interpreted as p(output | input, domain)?
>
> > - First, it is intutive to have the interpretation of the prompted language model as p(output | input, **prompt**), which follows the original work of GPT2 [2];
> >
> > - Second, **enforcing each prompt to be highly correlated to a domain** results in the interpretation of p(output | input, **domain**). Note that we maximize such correlation between a prompt and a domain via the agreement loss. The agreement loss constrains the backbone from making changes under previous prompts, which maximally enforces the hypernetwork to generate prompts that are strongly associated with domains. Otherwise, the current domain with a prompt misaligned to previous domains will be prevented from learning without full adaptation.
>
> ##### Q8: There is a lack of confidence interval and the quantitative values are quite close.
>
> > - The confidence intervals for most of the ablation experiment results are **smaller than 0.003**.
> > - For Table 2, the improvement contributed by each component is generally **above 0.004**, indicating determinism. Although the absolute value is small, it is considered a **significant improvement** in continual pre-training. For instance, the method [1] in ICLR2023 achieved a final improvement of 0.011.
> > - For Table 3, the values between several different loss metrics are indeed close. We have already mentioned in the paper that these metrics exhibit minimal differences, and we **chose a relatively more stable one**.
>
>
> [1] Zixuan Ke, et al. "Continual Pre-training of Language Models" ICLR 2023.
>
> [2] Alec Radford, et al. "Language Models are Unsupervised Multitask Learners" 2018.
>
> [3] Sen Lin, et al. “Theory on Forgetting and Generalization of Continual Learning” ICML 2023.
>
> [4] Samuel J. Bell, et al. “The Effect of Task Ordering in Continual Learning” Arxiv 2022.

---

### Meta-Review · Area_Chair_nNb1 · 2023-09-14

**Recommendation:** 3

**Metareview:**

The authors consider a continual learning setting without labels: models are trained in an unsupervised fashion given a sequence of unlabelled task inputs.
Evaluation happens via the introduction of labels: i.e., at any time in this training sequence, the model may be fine-tuned for any of the previous, current, or future tasks.
The proposed hypernetwork method outperforms alternatives on two datasets.

Reviewers largely agreed that this work is sound: experiments are thorough, baselines reasonable, etc. There was some discussion about continual fine-tuning vs. continual pretraining which was resolved. Existing limitations include that only 2 datasets and RoBERTa are considered but in response the authors address this by adding GPT2 and new experiments. Only classification is considered.

I personally wonder how much this proposed setting truly differs in practice from the general pre-training setting other than that the unlabelled text is provided on-line in task-sized chunks --- this setting also seems to be an extension of the somewhat common "unsupervised domain adaptation" setting where a pretrained model undergoes additional unsupervised training on a new unlabelled corpus (but here, there are a sequence of new domains, instead of just one).

Overall, it seems like the authors have addressed several concrete shortcomings (by adding new experiments on different models/datasets). The proposed method works better than a reasonable set of baselines. But, the importance of this setting beyond existing pretraning/unsupervised domain adaptation setups (both of which are closely related) is a bit less clear, which may be causing some reviewers to be less excited about this particular setup.

---

### Decision · Program_Chairs · 2023-10-07

**Decision:**

Accept-Findings

**Comment:**

The authors consider a continual learning setting without labels: models are trained in an unsupervised fashion given a sequence of unlabelled task inputs.
Evaluation happens via the introduction of labels: i.e., at any time in this training sequence, the model may be fine-tuned for any of the previous, current, or future tasks.
The proposed hypernetwork method outperforms alternatives on two datasets.

Reviewers largely agreed that this work is sound: experiments are thorough, baselines reasonable, etc. There was some discussion about continual fine-tuning vs. continual pretraining which was resolved. Existing limitations include that only 2 datasets and RoBERTa are considered but in response the authors address this by adding GPT2 and new experiments. Only classification is considered.

I personally wonder how much this proposed setting truly differs in practice from the general pre-training setting other than that the unlabelled text is provided on-line in task-sized chunks --- this setting also seems to be an extension of the somewhat common "unsupervised domain adaptation" setting where a pretrained model undergoes additional unsupervised training on a new unlabelled corpus (but here, there are a sequence of new domains, instead of just one).

Overall, it seems like the authors have addressed several concrete shortcomings (by adding new experiments on different models/datasets). The proposed method works better than a reasonable set of baselines. But, the importance of this setting beyond existing pretraning/unsupervised domain adaptation setups (both of which are closely related) is a bit less clear, which may be causing some reviewers to be less excited about this particular setup.